# A Novel PNDA-MMNet Model for Evaluating Dynamic Changes in the Brain State of Patients with PTSD During Neurofeedback Training

**DOI:** 10.3390/s25113522

**Published:** 2025-06-03

**Authors:** Peng Ding, Lei Zhao, Anmin Gong, Wenya Nan, Yunfa Fu

**Affiliations:** 1Faculty of Information Engineering and Automation, Kunming University of Science and Technology, Kunming 650500, China; ausar_schorr@stu.kust.edu.cn; 2Faculty of Science, Kunming University of Science and Technology, Kunming 650500, China; ynzhaolei@foxmail.com; 3School of Information Engineering, Chinese People’s Armed Police Force Engineering University, Xian 710086, China; gonganmincapf@163.com; 4Department of Psychology, College of Education, Shanghai Normal University, Shanghai 200234, China; wynan@shnu.edu.cn

**Keywords:** post-traumatic stress disorder, neurofeedback training, changes in brain state, mesoscopic network, process noise dynamic adaptation

## Abstract

Background: Monitoring and evaluating dynamic changes in brain states during electroencephalography (EEG) neurofeedback training (NFT) for post-traumatic stress disorder (PTSD) patients remains challenging when using traditional methods. Method: This study proposes a novel Process Noise Dynamic Adaptation-Mesoscale Mesonetwork Network (PNDA-MMNet) model, which improves upon conventional techniques by establishing a discrete linear dynamic model of the NFT process. The model utilizes a mesoscale intermediate network architecture to create a brain state observation matrix, computes the brain state transition matrix, and applies fuzzy rules for dynamic adaptive noise processing. This maximizes the separability between brain state transitions during NFT and resting states. Results: The proposed model achieves a brain state identification accuracy of 0.7428 ± 0.12 (area under the curve, AUC = 0.84), significantly outperforming conventional algorithms. Interpretations of the model indicate that continuous NFT reduces functional connectivity within the motor cortex, with stronger suppression in the right hemisphere compared to the left. Additionally, it reveals decreased activity in the occipital cortex, particularly in the left occipital region, where inhibition increases radially from the midline. Notably, the connectivity between the motor and occipital cortices remains stable throughout the training process. These connectivity changes reflect NFT-induced modulation of cortical activity and are consistent with known neurophysiological patterns in PTSD, highlighting their potential relevance to therapeutic mechanisms. Conclusion: This research introduces a more effective approach for real-time monitoring and evaluation of PTSD patients’ brain states during NFT, offering a quantitative method for assessing treatment efficacy and guiding therapeutic interventions.

## 1. Introduction

Post-traumatic stress disorder (PTSD) is a psychiatric condition that may develop after individuals are exposed to traumatic events such as natural disasters, serious illness, or violence [1]. An illustrative case is the 1994 Rwandan genocide [2], during which over one million people lost their lives [3]. Even more than 20 years later, the prevalence of PTSD among Rwandans aged 16 and older remains at 26.1%, and up to 41% among female survivors [4]. These data highlight the long-term psychological impact of mass trauma and the critical need for effective therapeutic interventions.

Among these, neurofeedback training (NFT) has emerged as a promising non-invasive technique for treating PTSD symptoms. For instance, Bois et al. demonstrated that EEG-based NFT can alleviate PTSD symptoms in genocide survivors, correlating improvements in clinical outcomes with increased resting-state alpha (8–12 Hz) band power [5,6,7]. However, most studies, including theirs, rely heavily on pre- and post-intervention comparisons to evaluate NFT efficacy [8,9,10,11,12].

Such pre–post assessments, while common, fail to capture the dynamic evolution of brain states during NFT sessions[13], thereby limiting understanding of the neural mechanisms driving symptom improvement. In clinical practice, this shortcoming reduces the possibility of adjusting NFT strategies in a timely and personalized manner.

To monitor ongoing brain changes during NFT, real-time indices such as Z-scores have been used [14]. These are derived from specific regions and frequency bands, converting raw power values into standardized deviations from normative means. Although this method offers temporal resolution, it is fundamentally limited in scope—capturing only individual signal deviations and offering little insight into the structure or evolution of global brain states [15]. Moreover, Z-score-based feedback does not account for inter-subject variability, reducing its interpretability and generalizability across patients and sessions.

Therefore, there is a critical need for a more comprehensive, interpretable, and individualized assessment framework for NFT, one that complements existing evaluation systems by tracking the evolving brain states of PTSD patients during training. This could both enhance clinical decision-making and contribute to uncovering the underlying mechanisms of NFT.

To address this need, the present study introduces a novel dynamic assessment approach for modeling brain changes during NFT in PTSD patients. Specifically, we propose a method called Process Noised Dynamic Adaptation-based Mesoscale Mesoscopic Network (PNDA-MMNet). This method establishes a discrete linear dynamic system in which the brain state is estimated and tracked during NFT. MMNet is used to construct the observation and transition matrices of the brain state, while PNDA adaptively adjusts process noise based on fuzzy rules, enhancing sensitivity to state changes.

Unlike classification-based or feature-point monitoring methods, PNDA-MMNet offers high interpretability by modeling the internal state transitions of the brain. We aim to provide a transparent mechanism for understanding how brain states evolve during NFT, moving beyond black-box evaluation.

Furthermore, the model is designed to extract consistent features across sessions and subjects, addressing the variability in NFT responses observed in clinical practice. This enhances its applicability for generalizing findings and refining training strategies across individuals.

In summary, this work contributes to NFT research by (1) proposing a dynamic brain state assessment method to improve mechanistic understanding of NFT; (2) developing an interpretable modeling framework rather than a simple classification tool; and (3) addressing cross-subject and cross-session variability by learning consistent representations of brain dynamics.

## 2. Materials and Methods

### 2.1. NFT Dataset for Patients with PTSD

The dataset used in this study includes a total of 26 PTSD patients, comprising data from 10 individuals found in publicly available datasets and 16 patients recruited through our experimental group and partner hospitals, all of whom received neurofeedback therapy. Each patient participated in between two and seven neurofeedback (NFT) sessions, during which they were instructed to regulate their brain activity by controlling a character in a visually presented video game paradigm [5].

For the 10 patient data obtained from the public dataset, detailed demographic and clinical information is available in reference [5] and is not repeated here to avoid redundancy(For more details, see the Appendix A). EEG data were acquired using two types of systems: (1) a 32-channel g.tec system, and (2) an 8-channel Flex EEG system.

For the remaining 16 patients recruited from our laboratory and collaborating clinical institutions, we collected additional demographic information and NFT session details. These data are summarized in Table 1 of the main manuscript, while a comprehensive description of diagnostic procedures, inclusion/exclusion criteria, and clinical assessments is provided in Section A.2 to maintain the conciseness of the main text. EEG recordings for these patients were obtained using a NeuSen W 32-channel EEG system (Neuracle Technology Co., Ltd., Shanghai, China).

Different EEG devices were selected to accommodate variations in head shape and recording environments.

To accommodate the different head shapes of the subjects, three types of EEG devices were used in this dataset:(1)g.tec device, featuring 32 electrodes in accordance with the international electrode placement standard. This device has an amplification factor of 20,000, employs an 8th order Butterworth filter (passband 0.5–100 Hz), 24-bit analog-to-digital (A/D) conversion, and a sampling rate of 250 Hz.(2)Flex EEG device, with 8 channels, including 3 bipolar and 5 unipolar channels [16]. The dataset obtained from this device comprises EEG channels from rows 1–8, specifically C3, Cz, C4, P3, O1, P7, Oz, and Pz, with row 9 being the label channel. This device has an amplification factor of 24, utilizes an 8th order Butterworth filter (passband 4–38 Hz), 16-bit A/D conversion, and a sampling rate of 250 Hz.(3)NeuSen W device, which also follows the 32-electrode layout, and uses a 250 Hz sampling rate.

For all EEG devices used, the reference electrode was consistently placed on the right earlobe, and the ground electrode was positioned at the AFz location. The dataset also includes experimental records for instances of invalid recordings and non-compliant experimental data.

### 2.2. NFT Process for Patients with PTSD

The following describes the paradigm procedure based on a publicly available dataset. Our own collected dataset follows the same paradigm and procedural flow.

The Neurofeedback Training for PTSD (NFT-PTSD) system utilizes the NeuroSensi gaming platform, developed by NeuroCONCISE Ltd. (Cambridge, UK) [17]. The procedural timeline is illustrated in Figure 1. Prior to and following the NFT sessions, patients with PTSD complete questionnaires on emotions and stress, which take approximately 5 min. Each session consists of 10 runs, with each run lasting 134 s. Before and after each run, there is a 134 s eyes-open resting and a 15 s rest period. During NFT, patients are tasked with controlling a game character (an astronaut) to maintain orbit in space, collecting rewards, and avoiding hazards.

Throughout the NFT-PTSD process, continuous EEG recordings are made from the Pz electrode (located above the midline parietal cortex), with band-pass filtering extracting the alpha rhythm (8–12 Hz). The power of this frequency band is calculated using a 500 ms sliding window, and patients control their game character’s orbital position by modulating this band’s power. In each run, the alpha band power threshold is set to less than 60% of the average alpha band power during the previous task (or resting state before the first task). When the real-time calculated average alpha band power for the current NFT task falls below this threshold, the game character is maintained in orbit; if the calculated band power exceeds the threshold, the NFT system issues commands causing the character controlled by the patient to deviate from the orbit. Patients achieve self-regulation of the alpha rhythm by real-time suppression of alpha band power throughout the NFT process.

This paradigm procedure forms the basis of both the publicly available data and our own data collection, ensuring consistency across datasets.

### 2.3. Linear Discrete Dynamic System Modeling for NFT Process in Patients with PTSD

This study models the NFT-PTSD patient process described in Section 2.2 as a linear discrete dynamic system [18], which can be represented by Equations (1) and (2):(1)$$x_{k}=A_{k}*x_{k-1}+B*u_{k}+w_{k-1}$$
(2)$$Z_{k}=H*x_{k}+v_{k}$$
where $x_{k}$ is the system state matrix [19], representing the observed matrix of the brain state at time k; $z_{k}$ is the observed value of the brain state [20], indicating the measurement signal of the brain state at time k; $A_{k}$ is the transition matrix [21], representing the matrix of brain state changes during the execution of the NFT task, used to quantify the impact of the NFT process on the brain state. We make the following assumptions about $A_{k}$:

**Assumption** **1.***In this dataset, each session consisting of 10 runs of NFT tasks is assumed to be identical. Therefore, we assume that* $A_{k}$ *is a constant matrix. Generalizing to a typical NFT system, if the impact of tasks on the brain state is time-varying, then*$A_{1}\neq A_{2}\neq \ldots \neq A_{k}$.

B is the control input matrix [22]. Since the NFT processes participated in by patients in this dataset are invariant, B is set to 0. Generalizing to a typical NFT system, such as an adaptive time-varying NFT system, B is not necessarily a constant zero matrix. $u_{k}$ is the control input value at time k [22]. When the NFT task changes at time k, this parameter quantifies the change to obtain the control input value. However, in this dataset, the NFT tasks involved are constant, so this term is set to 0. H is the observation transition matrix [23]. This matrix maps the brain state to the measurement signal. In this study, H is the inverse matrix of the lead matrix and the brain network computation, resulting in the EEG matrix. $w_{k-1}$ represents process noise [24], characterizing the fluctuations in brain state between the execution of the NFT task at time *k*−1 and the execution of the NFT task at time k. $v_{k}$ is measurement noise [25], representing the measurement noise of EEG in this experiment. k denotes the k th NFT task.

In summary, the system model of the NFT—PTSD patient process in this study can be further simplified to Equations (3) and (4).(3)$$x_{k}=A_{k-1}*x_{k-1}+w_{k-1}$$(4)$$z_{k}=H*x_{k}+v_{k}$$

In this model, the impact of the NFT process on the patient’s brain state is described using a state transition matrix. After adjusting and eliminating $w_{k-1}$, each row of the transition matrix $A_{k-1}$ is multiplied by each column of the observed matrix $x_{k-1}$ of the brain state during the k−1th NFT process. This operation yields the observed matrix $x_{k}$ of the brain state during the *k*-th NFT process. Therefore, each row of $x_{k-1}$ defines a linear combination, weighted by the columns of $A_{k-1}$, where the elements in $A_{k-1}$ serve as weights. The state transition values a_ij in $A_{k-1}$ correspond to the linear combination of the *j* th column of the observed matrix $x_{k-1}$ during the *k*−1 th NFT process and the *i* th row of the observed matrix $x_{k}$ during the *k* th NFT process.

In essence, we treat the process of neurofeedback as a step-by-step system where the patient’s brain state changes gradually over time. These changes are not random but follow a certain pattern influenced by the repeated NFT tasks. Control theory gives us a mathematical framework to describe such changes using matrices—essentially, structured tables of numbers that represent how different parts of the brain state influence each other. By assuming that the tasks remain consistent across sessions, we simplify the model to focus on the internal dynamics of the brain, which allows us to quantify how the training is working. The transition matrix in particular helps us identify how the brain responds from one task run to the next. This kind of modeling is commonly used in engineering but has rarely been applied in NFT research, making it a novel and promising direction for analyzing and interpreting brain state transitions.

### 2.4. NFT Data Alignment for Patients with PTSD

In the NFT—PTSD patient dataset, experimenters collected data using two different models of EEG devices (we clarify that three types of EEG devices were used in total: g.tec and Flex EEG in the public dataset, and NeuSen W in our collected dataset). To ensure comparability between the datasets, a common average reference method was employed in preprocessing to normalize the data from different devices to a set of commonly covered brain regions by the electrodes. Specifically, EEG data were resampled to 250 Hz, and a zero-phase 8th-order Butterworth bandpass filter (0.5–38 Hz) was applied using custom MATLAB (R2021b, MathWorks Inc., Natick, MA, USA) functions based on the butter and filtfilt operations.

Signals from channels C3, Cz, C4, P7, Pz, PO3, and Oz of the g.tec device were selected. For the Flex EEG device, signals from all channels were chosen, with the average signal of P3 and O1 being used as the “PO3” channel signal. From the NeuSen W EEG recordings, the same seven electrode positions were extracted to match the above configuration. The resultant common electrode coverage of brain regions, as shown in Figure 2, includes the left motor area, right motor area, left occipital area, and the midline parietal area.

To further minimize inter-device variability, all signals were re-referenced using the Common Average Reference (CAR) approach across the selected seven electrodes.

Subsequently, baseline correction was applied by subtracting the mean signal of the initial 0.5 s of each segment. Signals were then normalized using z-score normalization within each session to reduce amplitude-related inter-subject variability.

From the labels provided in the dataset, data for each patient’s sessions were obtained, each session consisting of 10 runs (trials) of NFT EEG segments. Invalid records and data from non-compliant experimental procedures were removed based on experimental records. Each valid run was extracted as a continuous 134 s segment, resulting in well-aligned and quality-controlled NFT and resting-state EEG data. This resulted in the acquisition of NFT and resting-state EEG data from 46 sessions, totaling 460 runs. Each run included 7 normalized common electrodes, with a duration of 134 s each. Each run included seven harmonized EEG channels (C3, Cz, C4, P7, Pz, PO3, Oz), preprocessed with the same standardized pipeline.

### 2.5. Process Noise Dynamic Adaptation Based Mesoscale Mesoscopic Network

Based on the linear discrete dynamic system model of the NFT-PTSD patient process established in Section 2.3, this study proposes a process noise dynamic adaptation based mesoscale mesoscopic network (PNDA-MMNet) method to assess the dynamic changes in brain state of patients with PTSD during NFT, as shown in Figure 3. In Figure 3a, the observed brain state quantity $z_{k}$ corresponds to the EEG data of the *k* th trial. The measurement noise $v_{k}$ within $z_{k}$ is removed through Independent Component Analysis (ICA), which was implemented in MATLAB using an automated algorithm developed in-house. ICA was applied in a model-driven manner rather than as a general preprocessing step, aiming to isolate and subtract non-neural noise components from $z_{k}$. Prior to ICA, EEG data were re-referenced using the common average reference (CAR) method. To preserve the full rank of the data matrix and avoid introducing ghost independent components, the original reference electrode was retained during re-referencing, following the recommendation of Kim H et al. [26].

Artifacts corresponding to ocular, muscular, or other non-neural sources were identified based on spatial and spectral features and automatically removed. After removal of $v_{k}$, the $z_{k}$ data were aligned to obtain commonly covered brain regions of aligned electrodes. The Spearman’s correlation coefficients are calculated between all electrodes of the aligned $z_{k}$ to obtain a Fully Connected Network (FCNet), where FCNet in this paper refers to connections between all recorded electrodes. The FCNet is divided into multiple sub-networks (Sub Network, SubNet) [27], including the Motor Cortex (MC) SubNet, the Left Occipital Cortex (LOC) SubNet, and the Motor Cortex-Left Occipital Cortex (MC-LOC) SubNet. Simultaneously, the commonly covered brain regions of aligned electrodes are categorized into areas between C3 and Cz (representing the Left Motor Cortex, LMC), between C4 and Cz (representing the Right Motor Cortex, RMC), and between P7, Pz, and Oz (representing the Left Occipital Cortex, LOC), as shown in Figure 3b,c. Following this categorization, the aligned $z_{k}$ is pooled, and from the pooled $z_{k}$, the Spearman’s correlation coefficients between different brain regions are calculated to form one mesoscale network (Middle Network, MNet). The brain state observation matrix $x_{k}$ is composed of MNet and multiple SubNets. A summary of these custom-defined brain region and network abbreviations is provided in Table 2 for reference.

In Figure 3d, the transition matrix $A_{k}$ is computed from the observed matrix $x_{k}$ of the brain state. $A_{k}$ characterizes the dynamic changes in the brain state of patients with PTSD during NFT. A Markov [28] chain is formed by 10 NFT processes (10 trials) in each session, and $A_{k}$ is computed for each session unit, where *k* ranges from 1 to 10. Following assumption 1, the covariance of $A_{1}$ to $A_{k}$ is calculated after linearizing $A_{k}$. Subsequently, using Equation (5), a loss function $f_{1}$ is defined, and the Markov chain is updated by iteratively minimizing the difference between $f_{1}$ before and after iteration. Process noise $w_{k-1}$ and state transition matrix $A_{k}$ are updated using Equations (6) and (3) until $A_{1}$ to $A_{k}$ are made as similar as possible.(5)$$f_{1}=\min[cov\left(A_{1},A_{2},\ldots ,A_{k}\right)]$$(6)$${\hat{w}}_{k-1}^{\left(j+1\right)}=\theta_{1}\left(\Delta f_{1}\right){\hat{w}}_{k-1}^{\left(j\right)}+b_{0}$$

In the equation, $\Delta f_{1}$ represents the difference between $f_{1}$ before and after updating the Markov chain, $\theta_{1}$ is the weight function for updating process noise $w_{k-1}$, ${\hat{w}}_{k-1}^{\left(j\right)}$ is the pre-update $w_{k-1}$, and ${\hat{w}}_{k-1}^{\left(j+1\right)}$ is the post-update $w_{k-1}$. $b_{0}$ is the bias value, and j indicates the number of updates.

In Figure 3e, the set of state transition matrices $A_{k}$ obtained from all sessions (including both self-collected and publicly available EEG data aligned via a unified NFT paradigm, as shown in Figure 3b) is divided into training, validation, and test sets in a ratio of 7:1:2. This partition was performed at the session level, ensuring no subject or session appeared in more than one set, thus preserving independence and preventing data leakage. Subsequently, a convolutional neural network (CNN) network consisting of 2 convolutional layers, 2 pooling layers, and 2 fully connected layers was employed to classify $A_{k}$ corresponding to NFT processes and resting. Finally, using Equation (7), the loss function $f_{2}$ is calculated to determine $\Delta f_{2}$, updating $b_{0}$ for each session. $b_{0}$ is utilized to align the feature spaces of different sessions.(7)$$f_{2}=\min\left[dist\left(\sum_{i=1}^{46}A_{1,i},A_{2,i},\ldots ,A_{k,i}\right)\right],k=1,\;2,\;\ldots ,\;10$$(8)$${\hat{b}}_{0}^{\left(j+1\right)}=\theta_{2}\left(\Delta f_{2}\right){\hat{b}}_{0}^{\left(j\right)}$$

In the equation, $A_{k,i}$ represents the *k* th state transition matrix for the i th session, $\Delta f_{2}$ is the difference between $f_{2}$ before and after updating $b_{0}$, $\theta_{2}$ is the weight function for updating $b_{0}$, ${\hat{b}}_{0}^{\left(j\right)}$ is the pre-update $b_{0}$, and ${\hat{b}}_{0}^{\left(j+1\right)}$ is the post-update $b_{0}$. Here, *j* denotes the number of updates.

### 2.6. Statistical Assessment of Brain State Transition Weights Between NFT and Resting States with FDR Correction

To test the hypothesis that brain state transition weights and decoding performance metrics significantly differ between NFT and resting periods (or between different models), we applied two-tailed independent-sample *t*-tests. The assumptions of normality and homogeneity of variance were assessed using Shapiro–Wilk and Levene’s tests, respectively. In cases of assumption violation, non-parametric alternatives such as the Mann–Whitney U test were considered but ultimately not required. Multiple comparisons were corrected using the Benjamini–Hochberg procedure to control for false discovery rate (FDR), with statistical significance set at q < 0.05, where q represents the FDR-adjusted *p*-value.

In addition, we evaluated differences in performance metrics (e.g., accuracy, true negative rate [TNR], etc.) across various machine learning and deep learning models. Statistical comparisons were performed to identify significant differences in these parameters, providing a comprehensive understanding of model performance.

## 3. Results

### 3.1. Full Connectivity Networks of Patients with PTSD During NFT and Resting

Figure 4a presents a scatter plot visualizing the t-SNE dimensionality reduction in original EEG data from 46 Sessions during NFT and resting periods. There was no statistical difference between the samples from the two periods (q > 0.05, FDR-corrected). Figure 4b displays a scatter plot showing the linearized EEG brain networks calculated during the NFT and resting periods of the 46 sessions, again reduced in dimensionality by t-distributed stochastic neighbor embedding (t-SNE) [29]. Similarly to the previous result, there was no statistical difference between the samples from these two periods (*p* > 0.05). Figure 4c,d illustrate the brain networks of a PTSD patient during the same run in a single Session, comparing NFT and resting periods. Significant differences were observed in some of the brain region connections. A comprehensive statistical analysis indicated that 46.09% of the runs exhibited statistically significant differences (q < 0.05, FDR-corrected) in brain networks between NFT and resting periods across all connections. Furthermore, 96.74% of the runs showed significant differences (*p* < 0.05) in brain networks between the NFT and resting periods in 75% of the connections.

### 3.2. Classification Results of Brain State During NFT and Resting

The PNDA-MMNet proposed in this study achieved an average accuracy of 0.7428 (±0.12) in classifying brain state during NFT and resting. This performance showed significant differences (q < 0.05, FDR-corrected) compared to the average classification accuracies obtained with CNN, long short-term memory (LSTM) recurrent neural networks, ensemble learning (ENS; where bagging is based on decision trees, and subspace is based on Linear Discriminant Analysis, LDA), and the traditional K-Nearest Neighbors(KNN) method [30,31,32], All parameters of the comparison models are detailed in Table A1 of Section A.1. Methods with a lower accuracy in comparative studies are primarily due to the heterogeneity exhibited by EEG signals across subjects and sessions. However, the approach proposed in this study adequately addresses this issue within its model.

Although some recent studies have explored Graph Neural Networks (GNN) [32], transformer-based architectures [33], and hybrid attention mechanisms [34] for EEG decoding, these models often require large-scale data and stable signal distributions to perform optimally. In our setting, characterized by relatively short time sequences, high inter-subject/session variability, and limited training samples, preliminary experiments showed that such models underperformed—failing to converge stably or yielding lower accuracy than LSTM-based approaches.

Specifically, the average classification accuracy of the GNN model was around 0.35(±0.11), and that of the transformer-based model was approximately 0.57(±0.08), both substantially lower than the 74.28% achieved by PNDA-MMNet.

In addition to these practical constraints, our model design was also guided by the conceptual goal of process evaluation rather than clinical outcome prediction. That is, we aim to decode and interpret brain state transitions during the course of neurofeedback training, rather than treating classification accuracy as a direct proxy for therapeutic benefit. From this perspective, interpretability and stability were prioritized over marginal performance gains.

Therefore, we selected representative and practically effective architectures as baselines, which remain widely used and benchmarked in the field. The proposed method also showed significant advantages in terms of the balanced F1-Score and True Positive Rate (TPR) compared with other methods (q < 0.05, FDR-corrected). Although the True Negative Rate (TNR) achieved with the proposed method was lower than that of LSTM, it was higher compared to the other methods, as illustrated in Table 3 and Figure 5.

### 3.3. Statistical Differences Between Transition Matrix for Brain State During NFT and Resting

The observation matrix for brain state $x_{k}$ for patients with PTSD consists of four EEG networks (three SubNets and one MNet), representing the dynamic changes in brain state of patients with PTSD during NFT. The corresponding brain state transition weight matrix $A_{k}$ is also composed of four parts, as shown in Figure 6a,b, where $A_{k}$(MC), $A_{k}$(LOC), $A_{k}$(MC-LOC), and $A_{k}$(MNet) are calculated from the MC SubNet, LOC SubNet, MC-LOC SubNet, and MNet, respectively. The weight between C3 and C3 in $A_{k}$ represents the coefficient of the transformation of the connection value in row C3 of $x_{k-1}$ to the connection value in column C3 of $x_{k}$. In all runs, there were statistically significant differences in the weights between C3 and C3 (*p* = 0.012), Cz and Cz (*p* = 0.012), and C4 and C4 (*p* = 0.0180) in $A_{k}$(MC) during NFT and Resting periods, after FDR correction (q < 0.05), while the weights between all channels in $A_{k}$(MC-LOC) showed no statistical difference. In $A_{k}$(LOC), except for the weight between PO3 and Pz, weights between other channels showed significant differences (q < 0.05, FDR-corrected), as illustrated in Figure 6a. In all runs, the weights between LMC and LMC, RMC and RMC, and LOC and LOC in $A_{k}$(MNet) during NFT and resting periods showed statistical differences (q < 0.05, FDR-corrected), as shown in Figure 6b. Figure 6c presents a scatter plot visualizing the process noise $w_{k-1}$ from all runs during NFT and resting periods, linearized and reduced in dimensionality by t-SNE, showing no separability between $w_{k-1}$ during NFT and resting periods.

Further calculations of the ratio of statistically significant weights between channels in $A_{k}$ during NFT and resting periods are shown in Figure 6d–f, indicating that the weights in $A_{k}$ during NFT are consistently lower than during resting periods.

## 4. Discussion

BCI-based NFT has been demonstrated to provide a cost-effective intervention method for patients with PTSD. The evaluation of NFT is crucial for further research and application of this approach. However, existing pre–post assessments fail to monitor the changes in the brain state of patients with PTSD during the NFT process. To address this issue to some extent, this study introduced the PNDA-MMNet method. This approach models the NFT process as a discrete linear dynamic system and then designs a method to adaptively adjust process noise to solve for the transition matrix for brain state during NFT, revealing the impact of NFT on the brain.

### 4.1. Robustness of PNDA-MMNet to EEG Heterogeneity and Limited Spatial Resolution

In terms of classification accuracy and model parameter evaluation metrics for the brain state transition matrices during both NFT and resting processes, our method outperformed ENS, CNN, LSTM, and KNN. From a system modeling perspective, this may be due to not directly using the observation matrix as an input for the classification model, but instead, by adjusting and eliminating process noise, smoothing out heterogeneity across different runs. Moreover, the results in Figure 6c show that the eliminated $w_{k-1}$ from the linear discrete dynamic system, after visualization, displayed a heterogeneity similar to the original EEG data (Figure 4a), indirectly validating the effectiveness of the proposed method for eliminating process noise.

Importantly, although the use of 8-channel Flex EEG may limit the spatial granularity of brain network analysis, we addressed this constraint by designing PNDA-MMNet to operate effectively under low-density EEG conditions. The model incorporates both localized and system-level information within a unified dynamic framework, allowing for mesoscopic-level representations of neural state transitions. This design helps mitigate the reduced spatial coverage while maintaining sufficient sensitivity to detect NFT-induced changes in brain dynamics.

Compared to the approach presented in this paper, existing studies [8,9,10,11,12] using pre-post NFT comparisons, including brain signal features and clinical symptom comparisons, are important methods for evaluating NFT. However, they struggle to eliminate the heterogeneity introduced by the experiment. By contrast, our method not only improves dynamic modeling accuracy but also enhances robustness against data variability, even under constrained EEG configurations.

Notably, to evaluate generalization ability, we combined self-collected and public datasets with aligned NFT paradigms, and randomly split all sessions into training, validation, and test sets in a 7:1:2 ratio. This ensured cross-subject, cross-session, and cross-source variability, and avoided data leakage between subsets.

### 4.2. Model Interpretability and Mapping to Neurophysiology

To address the interpretability of the PNDA-MMNet model, we explicitly link each component of the model to neurophysiological constructs derived from prior knowledge of PTSD-related brain dysfunction. First, the observed state matrix ($x_{k}$), derived from EEG recordings, integrates functional connectivity metrics across anatomically defined regions—including the left and right motor cortices (LMC, RMC) and the lateral occipital cortex (LOC)—through Spearman’s correlation coefficients within and between these subnetworks (MC, LOC, and MC-LOC). These network-level observations form the inputs to the dynamic state-space model, enabling physiologically meaningful representation of brain state evolution.

The state transition matrix ($A_{k}$), computed at each NFT trial within a session, quantifies the directional change in brain states and is interpreted as capturing how NFT modulates inter-regional connectivity and activity. For instance, the progressive suppression of connectivity within C3, Cz, and C4 channels during NFT trials reflects reduced activity in the motor cortex, consistent with therapeutic mechanisms aiming to mitigate hyperarousal symptoms in PTSD. Similarly, the LOC subnetwork shows decreasing connectivity from medial to lateral regions, capturing spatially specific occipital deactivation.

The process noise term ($w_{k-1}$), dynamically adapted via Markov chain optimization, reflects trial-to-trial fluctuations in brain states not directly explained by prior transitions. These may be interpreted as capturing individual variability in engagement or responsiveness to NFT. Finally, the CNN-based classification layer interprets the optimized transition matrices ($A_{k}$) to distinguish NFT from resting states, effectively mapping dynamic neurophysiological patterns to cognitive-behavioral states.

Taken together, PNDA-MMNet is not merely a predictive tool but a computational framework that encodes, monitors, and interprets NFT-induced modulation of motor and occipital cortical networks—aligning with established neurobiological models of PTSD.

### 4.3. Dynamic Brain State Changes in PTSD During NFT Revealed by PNDA-MMNet

The features extracted by the proposed PNDA-MMNet model show separability between the NFT and resting processes for the following reasons: in the NFT process, the weights between channels C3 and C3, Cz and Cz, C4 and C4 in $A_{k-1}$ were, respectively, 47.71%, 40.25%, and 13.72% of their corresponding weights during the resting process. As shown in Figure 7, this indicates a continual weakening of connectivity between C3, Cz, and C4 during NFT, effectively suppressing overall activity in the Motor Cortex. Additionally, the suppression ratio of the weight between C4 and C4 channels in $A_{k-1}$ was greater than that between C3 and C3, suggesting a higher degree of suppression in the Right Motor Cortex compared to the Left Motor Cortex. Similarly, NFT also inhibited the overall activity in the Lateral Occipital Cortex, with greater suppression in the Oz and P7 regions compared to the PO3 region, and more in the PO3 region than in the Pz region. This suggests that the degree of inhibition in the LOC area during NFT increases radially from the midline toward the periphery of the occipital region. Furthermore, there was no statistical difference between $A_{k}$(MC-LOC) during NFT and resting processes, indicating that NFT did not alter the connectivity between MC and LOC. Statistical analysis of $A_{k}$(MNet) showed that NFT suppressed the activity in LMC, RMC, and LOC, with the greatest suppression in RMC, followed by LMC. This is consistent with the results from the statistical analysis of $A_{k}$(MC) and $A_{k}$(LOC). However, existing pre-post NFT assessment methods cannot monitor the changes in brain state of patients during NFT, whereas the proposed PNDA-MMNet model can assess the dynamic changes in brain state of patients with PTSD during NFT.

These results align with established neurobiological models of PTSD, which implicate dysregulation in motor and occipital cortices related to hyperarousal and visual processing abnormalities [35,36]. The suppression observed in these regions during NFT suggests that the PNDA-MMNet model captures key neurophysiological changes relevant to PTSD symptomatology [37], offering a computational perspective on NFT’s potential mechanism of action.

### 4.4. Limitation

A limitation of this study is that the PNDA-MMNet model, while effective in monitoring brain state transitions during NFT, may oversimplify the complex and individualized nature of neural processes in PTSD. The model assumes a discrete linear dynamic system, which may not capture all nonlinearities in brain activity. Additionally, its generalizability to larger and more diverse patient populations needs further validation. While the current model achieves reasonable inference times on small datasets (on the order of 10 s), its complexity may hinder real-time performance in practical neurofeedback systems, where near-instantaneous response is critical. Therefore, further optimization is necessary to enhance its suitability for clinical deployment.

## 5. Conclusions

Compared to existing pre–post NFT assessment methods, this paper introduces a novel PNDA-MMNet model for characterizing and evaluating the dynamic changes in the brain state of patients with PTSD during NFT. The results of brain state recognition during NFT and resting, as well as the statistical results of brain regions contributing significantly to classification, demonstrate the effectiveness of the proposed model. This study aims to provide insights into the quantitative assessment of the changes in brain state of patients undergoing NFT.

## Figures and Tables

**Figure 1 sensors-25-03522-f001:**
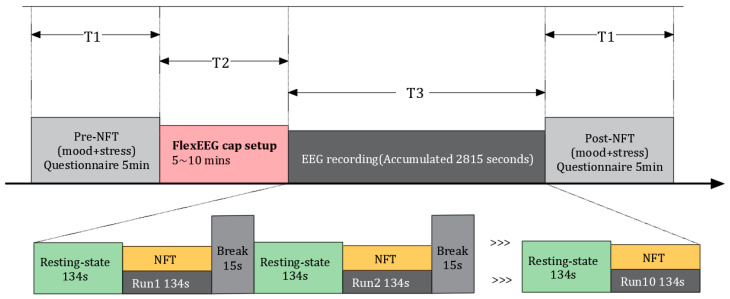
Timing for the NFT PTSD process. The symbol “>>>“ indicates Run3 to Run9 in the figure.

**Figure 2 sensors-25-03522-f002:**
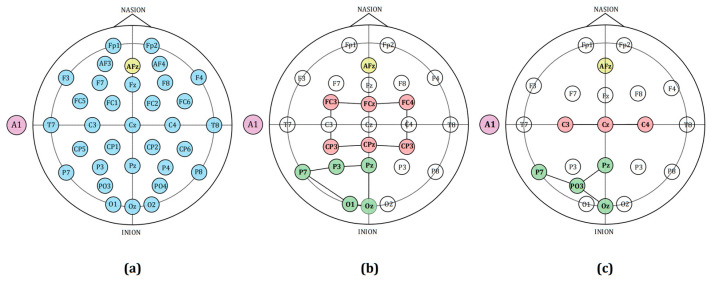
Channel selection and normalized common electrode coverage for g.tec and Flex EEG Devices. (**a**) Channels recorded using the g.tec device, marked in blue; (**b**) bipolar and unipolar channels of the Flex EEG device marked in red and green, respectively; (**c**) normalized common electrode coverage areas, with channels covering the left and right motor areas marked in red, and channels covering the left occipital area marked in green. In the figure, the reference electrodes are marked in purple and the ground electrodes in yellow.

**Figure 3 sensors-25-03522-f003:**
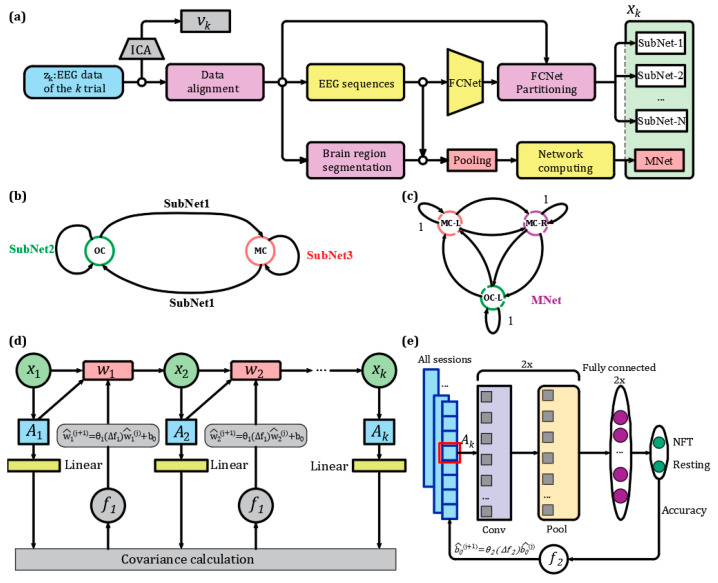
The methods for PNDA-MMNet. (**a**) Joint representation of based on sub-networks and mesoscale networks; (**b**) three sub-networks within and between MC and OC; (**c**) mesoscale network composed of correlations between LMC, RMC, and LOC; (**d**) dynamic adaptation of process noise $w_{k-1}$ and computation of state transition matrix $A_{k}$; (**e**) classification of $A_{k}$ during NFT and resting using a CNN and adjustment of $b_{0}$. In the figure, MC represents the motor cortex area, and OC represents the occipital cortex area; LMC is the region between C3 and Cz; RMC is the region between C4 and Cz; LOC is the region between P7, Pz and Oz. NFT and resting represent the NFT process and resting, respectively. Conv stands for the convolutional layer, and pool stands for the pooling layer.

**Figure 4 sensors-25-03522-f004:**
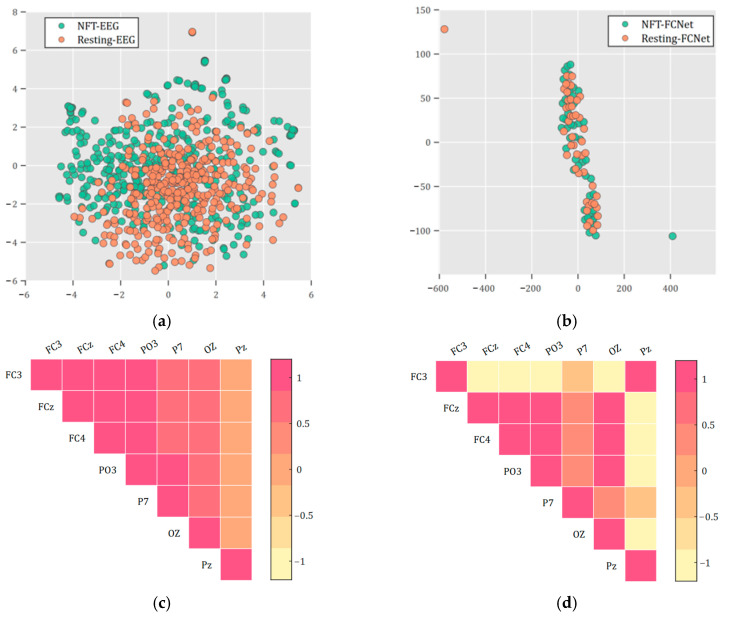
Distribution of original samples and FCNet for patients with PTSD during NFT and resting: (**a**) scatter plot of original EEG data during NFT and resting; (**b**) scatter plot of brain network features during NFT and resting; (**c**) brain network of a patient with PTSD during a single run in the same session for NFT; (**d**) brain network of a patient with PTSD during a single run in the same session for resting. In this study, all connections between recorded electrodes are referred to as a FCNet. NFT-EEG represents EEG during NFT, and Resting-EEG represents EEG during resting. Similarly, NFT-FCNet represents the full connectivity network during NFT, and Resting-FCNet represents the full connectivity network during resting.

**Figure 5 sensors-25-03522-f005:**
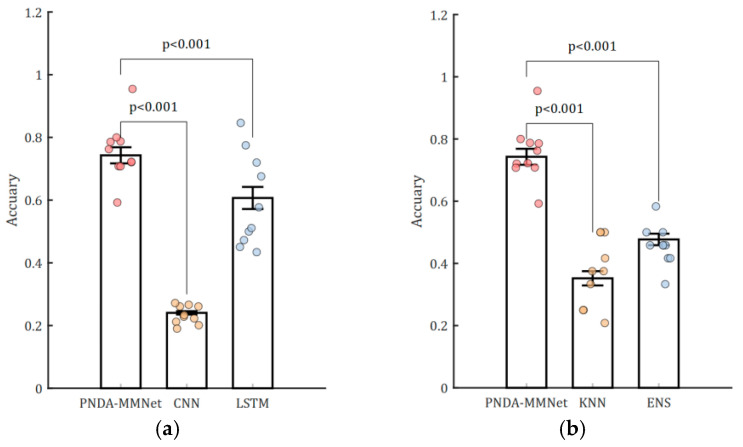
Statistical testing for classification results of brain state during NFT and resting. (**a**) Statistical testing for classification results by PNDA-MMNet, CNN, and LSTM; (**b**) statistical testing for classification results by PNDA-MMNet, KNN, and ENS. A subset (30%) of data points are randomly selected and overlaid as scatter points on the bar graphs with error bars to avoid overcrowding and improve clarity.

**Figure 6 sensors-25-03522-f006:**
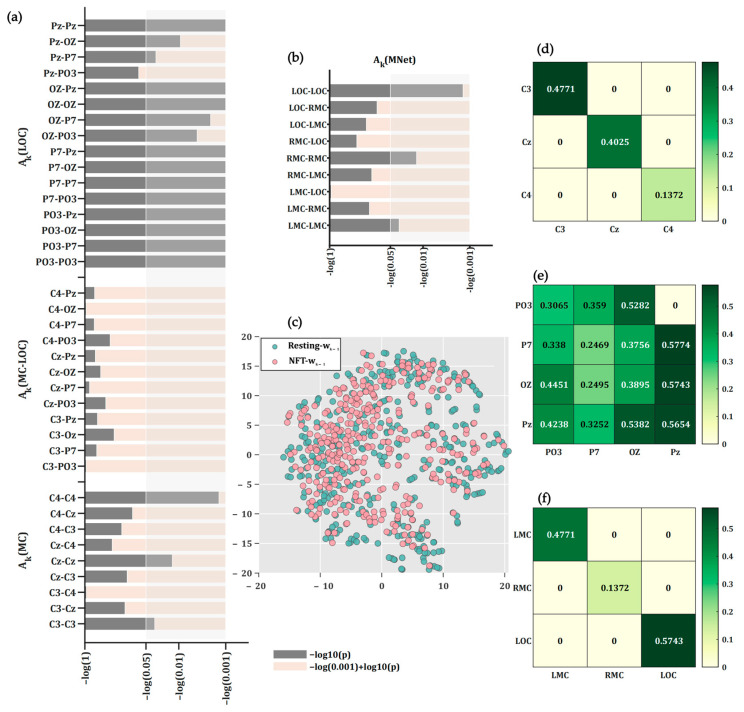
Statistical results of transition matrices for brain state $A_{k}$ during NFT and resting. (**a**) Statistical results of $A_{k}$ for the three SubNets, where the horizontal axis in the figure represents the logarithm of *p*-values, followed by negation; (**b**) statistical results of $A_{k}$ for Mnet; (**c**) scatter plot of $w_{k-1}$ during NFT and resting; (**d**) ratio of statistically significant weights between channels in $A_{k}$ calculated by MC SubNet during NFT and resting; (**e**) ratio of statistically significant weights between channels in $A_{k}$ calculated by LOC SubNet during NFT and resting; (**f**) ratio of statistically significant weights between channels in $A_{k}$ calculated by MNet during NFT and resting. Note: MC refers to the region between C3 and C4; LMC refers to the region between C3 and Cz; RMC refers to the region between C4 and Cz; LOC refers to the region between P7 and Pz, and Oz; *A* weight ratio of 0 between channels in $A_{k}$ indicates no statistical difference in weights between the channels.

**Figure 7 sensors-25-03522-f007:**
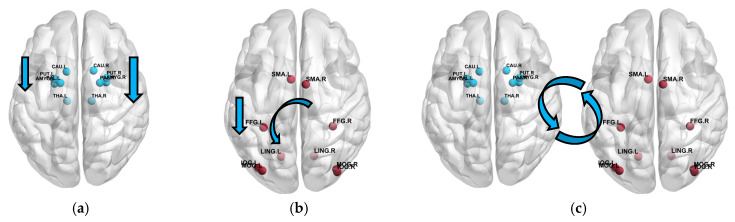
Explaining the mechanism of NFT on the brain state of PTSD patients based on PNDA MMNet method. (**a**) $A_{k}$ (MC); (**b**) $A_{k}$( LOC); (**c**) $A_{k}$(MC—LOC)).

**Table 1 sensors-25-03522-t001:** Demographic and clinical information of the 16 in-house PTSD patients.

Subject ID	Gender	Age (Years)	No. of NFT Sessions	Stressor
S01	FM	57	3	Car accident, Physical assault
S02	M	50	2	Bereavement
S03	FM	19	5	Sexual assault, School bullying
S04	FM	62	3	Bereavement
S05	M	14	4	School bullying
S06	FM	61	2	Natural disaster
S07	FM	18	4	Bullying
S08	FM	20	4	Bullying
S09	FM	29	2	Romantic breakup
S10	FM	18	2	Domestic violence
S11	FM	13	7	School bullying
S12	M	14	2	School bullying
S13	M	25	3	Romantic breakup
S14	FM	33	4	Workplace bullying
S15	M	16	5	Domestic violence
S16	FM	47	2	Natural disaster, Bereavement

Note: M = male; FM = female.

**Table 2 sensors-25-03522-t002:** Glossary of brain region and network abbreviations.

Abbreviation	Full Term	Description
FCNet	Fully Connected Network	Network comprising correlations between all EEG channels
SubNet	Subnetwork	Defined subcomponents of FCNet by anatomical/functional regions
MC-SubNet	Motor Cortex Subnetwork	Includes electrodes corresponding to motor areas
LOC-SubNet	Left Occipital Cortex Subnetwork	Includes electrodes over the left occipital region
MC–LOC-SubNet	Motor Cortex–Left Occipital Cortex Subnetwork	Represents inter-region connectivity between motor and occipital areas
MNet	Mesoscale Network	Network constructed from region-wise averaged correlations
LMC	Left Motor Cortex	\
RMC	Right Motor Cortex	\
LOC	Left Occipital Cortex	\

Note: While relevant abbreviations are defined upon first use in the text, this table is included to improve readability in subsequent methodological and analytical descriptions, especially where multiple custom-defined terms are used.

**Table 3 sensors-25-03522-t003:** Classification results of brain state during NFT and resting.

Method	Accuracy (SD)	F1-Score (SD)	TPR (SD)	TNR (SD)
CNN	0.2405 (0.02)	0.3241 (0.02)	0.5500 (0.51)	0.4500 (0.51)
LSTM	0.6069 (0.16)	0.5936 (0.016)	0.4561 (0.41)	0.7543 (0.29)
ENS	0.4771 (0.08)	0.4799 (0.09)	0.4527 (0.08)	0.5067 (0.15)
KNN	0.3521 (0.10)	0.3457 (0.11)	0.3073 (0.12)	0.3621 (0.17)
**PNDA-MMNet**	**0.7428** **(0.12)**	**0.7302** **(0.11)**	**0.8477 (0.14)**	0.5912 (0.17)

The advantages of this model over classical models are highlighted in bold.

## Data Availability

Data are contained within the article and Appendix A.

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
