# Peer review of "A Novel PNDA-MMNet Model for Evaluating Dynamic Changes in the Brain State of Patients with PTSD During Neurofeedback Training"

_sensors, 2025, doi:10.3390/s25113522_

Round 1
Reviewer 1 Report
Comments and Suggestions for Authors
This study proposed a new method called PNDA-MMNet model to analyze real-time brain state changes during neurofeedback training for PTSD patients. Their proposed method achieved better classification performance than conventional algorithms. The model suggested suppression in motor and occipital cortex activity, especially in the right hemisphere, during NFT, which aligns with existing PTSD models. I believe the findings would contribute to the field, but the manuscript needs some improvement.
The introduction gives general background on PTSD and neurofeedback training, but the focus of the study is not made clear early on. Too much space is spent discussing the Rwandan genocide, which, while informative, feels disconnected from the main technical objective. The authors mention the limitations of pre-post assessments and briefly introduce Z-score methods, but the discussion lacks depth. They don’t clearly explain why current monitoring methods are insufficient or what specific problem their model is solving. The new method is introduced at the end, but without a clear buildup. There is no research question, no hypothesis, and no strong justification for the proposed approach. Overall, the introduction reads more like a collection of background facts than a structured setup for the study.
The description of participants is incomplete. The authors stated that the dataset includes PTSD patients from public datasets and hospitals but provide no detail about clinical characteristics, diagnostic criteria, demographics, or inclusion/exclusion criteria. Without this information, it is difficult to interpret the generalizability of the findings or assess whether the sample is representative. I think this is particularly important for a study aiming to evaluate therapeutic mechanisms in PTSD.
Technical details are missing:
List all EEG channels.
The paper lacks a clear description of EEG preprocessing beyond hardware-level filtering. There is no mention of standard steps such as baseline correction, artifact rejection (other than vague ICA use), epoching, or signal normalization. For a study involving multichannel EEG and network analysis, a transparent preprocessing pipeline is essential. Currently, all the descriptions are vague. The authors must detail the preprocessing procedures step by step, in chronological order, without any omissions, and include all relevant parameters and tools used. Typically, EEG preprocessing should be described in a dedicated, separate section in the methods.
The use of ICA is poorly described. The authors mention that ICA is applied to remove measurement noise from EEG data, but do not specify how ICA was implemented, which components were removed, or whether artifact rejection was done manually or automatically. Without this information, it is unclear what role ICA plays in the preprocessing pipeline and whether it might affect the downstream analysis. More methodological detail is needed.
The authors applied ICA as part of their preprocessing but did not explain how average referencing was handled. According to [1], incorrect average referencing can cause effective rank deficiency and introduce ghost ICs, which directly impacts ICA decomposition quality. The authors should confirm whether average referencing was applied correctly and whether the input data were effectively rank-full prior to ICA. Otherwise, ghost components may be introduced, compromising the entire analysis. If the initial reference was included, simply stating “We included the initial reference for re-referencing to the average, keeping the data rank full [1]” would be fine. If preprocessing was done in a way that preserved rank, this should be made explicit by stating something along the lines of what I mentioned above. Please refer to the paper and engage with it directly.
[1] https://doi.org/10.3389/frsip.2023.1064138
The paper lacks a description of the statistical analysis plan, including how the methods were chosen to test the hypotheses. A proper statistical approach should be designed to evaluate whether the observed results support the authors' claims, not just report descriptive differences. The methods section should specify the type of tests used, whether the tests were directional, what assumptions were checked, and how the outcomes would be interpreted in favor or against the hypotheses.
The authors compare several models, but make no mention of multiple comparison correction. This is a basic statistical requirement when evaluating several classifiers, especially when claiming performance differences.
The discussion feels overly shallow. It mostly restates the results without critically interpreting them or linking them back to the broader literature. There is little reflection on how well the findings support the original aims, and no serious consideration of limitations or alternative explanations. Main claims are left ungrounded, and the discussion does not add much beyond what is already apparent in the results. This section needs to be expanded to offer real insight and connect the findings to existing knowledge.
Author Response
Response to Reviewer 1 Comments
Summary:We sincerely appreciate your valuable comments and the time you devoted to reviewing our manuscript. We have carefully addressed each of your suggestions and revised the manuscript accordingly. All changes are highlighted in red in the revised version. We hope that our responses and modifications will meet your approval.
Comment 1:
The introduction gives general background on PTSD and neurofeedback training, but the focus of the study is not made clear early on. Too much space is spent discussing the Rwandan genocide, which, while informative, feels disconnected from the main technical objective. The authors mention the limitations of pre-post assessments and briefly introduce Z-score methods, but the discussion lacks depth. They don’t clearly explain why current monitoring methods are insufficient or what specific problem their model is solving. The new method is introduced at the end, but without a clear buildup. There is no research question, no hypothesis, and no strong justification for the proposed approach. Overall, the introduction reads more like a collection of background facts than a structured setup for the study.
Reply to Comment 1:
We sincerely thank the reviewer for pointing out the lack of clarity and focus in the original Introduction. We agree that the prior version spent disproportionate space on background details (e.g., the Rwandan genocide) without clearly building up the research question or highlighting the technical innovation.
In the revised manuscript, we have substantially rewritten the Introduction to improve its structure and focus. Specifically:
We streamlined the description of the Rwandan context, using it only to motivate the relevance of PTSD-related NFT interventions.
We clearly stated the research gap in current NFT evaluation methods—namely, the insufficiency of pre-post comparisons and the limitations of real-time Z-score monitoring.
We introduced our proposed model (PNDA-MMNet) earlier, clarifying its objective to track dynamic brain state changes during NFT with high interpretability.
We also explicitly articulated the three core contributions of our work: (1) proposing a new evaluation framework for brain dynamics during NFT, (2) providing an interpretable modeling approach, and (3) addressing cross-subject/session variability by extracting consistent features.
These revisions aim to ensure that the reader understands the motivation, research question, and contribution of our work from the outset. The updated Introduction now serves as a clearer and more direct setup for the technical sections that follow.
The revised text appears in the introduction section of the marked manuscript, highlighted in red.
Comment 2:
The description of participants is incomplete. The authors stated that the dataset includes PTSD patients from public datasets and hospitals but provide no detail about clinical characteristics, diagnostic criteria, demographics, or inclusion/exclusion criteria. Without this information, it is difficult to interpret the generalizability of the findings or assess whether the sample is representative. I think this is particularly important for a study aiming to evaluate therapeutic mechanisms in PTSD.
Reply to Comment 2:
We thank the reviewer for the insightful comment regarding the incomplete description of the participants. In response, we have made the following revisions:
For the PTSD patient cohort, we have now provided detailed information regarding the inclusion and exclusion criteria, as well as additional clinical and demographic data for the 16 patients recruited from our experimental group and partner hospitals. This information has been summarized in Table 1 of the manuscript to ensure clarity while maintaining readability.
In Appendix A2, we have included a comprehensive description of the inclusion/exclusion criteria, clinical assessments, and demographic characteristics of the participants, in addition to details of the clinical measures used (e.g., CAPS, PCL-5, HAMD). This allows the reader to assess the generalizability and representativeness of our sample, while keeping the main text concise.
For the 10 PTSD patients from the public dataset, we have clarified that detailed demographic and clinical information is available in Reference [5] and is not repeated in the main text to avoid redundancy.
These revisions address the concerns about the generalizability and representativeness of our sample, and we hope the provided details now meet the reviewer's expectations.
Thank you for your valuable feedback, which has significantly improved the transparency and rigor of our manuscript.
The revised text appears in the section 2.A and Appendix A2 of the marked manuscript, highlighted in red.
Comment 3:
Technical details are missing:
List all EEG channels.
The paper lacks a clear description of EEG preprocessing beyond hardware-level filtering. There is no mention of standard steps such as baseline correction, artifact rejection (other than vague ICA use), epoching, or signal normalization. For a study involving multichannel EEG and network analysis, a transparent preprocessing pipeline is essential. Currently, all the descriptions are vague. The authors must detail the preprocessing procedures step by step, in chronological order, without any omissions, and include all relevant parameters and tools used. Typically, EEG preprocessing should be described in a dedicated, separate section in the methods.
Reply to Comment 3:
We sincerely thank the reviewer for this insightful and constructive comment. In response, we have substantially revised and expanded the EEG preprocessing section to clearly describe the complete data processing pipeline in a step-by-step and chronological manner. The updated content now includes the following detailed procedures:
Channel selection and harmonization across the three EEG devices (g.tec, Flex EEG, and NeuSen W), ensuring the use of seven common electrodes: C3, Cz, C4, P7, Pz, PO3, and Oz.
(1) Resampling of all EEG data to 250 Hz and zero-phase 8th-order Butterworth bandpass filtering (0.5–38 Hz), using MATLAB’s butter and filtfilt functions.
(2) Application of a Common Average Reference (CAR) method for re-referencing.
(3) Baseline correction using the initial 0.5 seconds of each segment.
(4) Z-score normalization within each session to reduce amplitude-based variability across subjects and devices.
(5) Epoch extraction, where each valid run (trial) is retained as a continuous 134-second segment, after removing invalid or non-compliant trials based on log records.
This detailed preprocessing pipeline is now presented as a clearly demarcated part of the manuscript under the revised subsection “D. NFT Data Alignment for Patients with PTSD” (Methods Section). All relevant parameters and tools are now explicitly mentioned. We believe this improvement enhances the reproducibility and transparency of our methodology, and we thank the reviewer again for prompting us to make this important clarification.
Comment 4:
The use of ICA is poorly described. The authors mention that ICA is applied to remove measurement noise from EEG data, but do not specify how ICA was implemented, which components were removed, or whether artifact rejection was done manually or automatically. Without this information, it is unclear what role ICA plays in the preprocessing pipeline and whether it might affect the downstream analysis. More methodological detail is needed.
The authors applied ICA as part of their preprocessing but did not explain how average referencing was handled. According to [1], incorrect average referencing can cause effective rank deficiency and introduce ghost ICs, which directly impacts ICA decomposition quality. The authors should confirm whether average referencing was applied correctly and whether the input data were effectively rank-full prior to ICA. Otherwise, ghost components may be introduced, compromising the entire analysis. If the initial reference was included, simply stating “We included the initial reference for re-referencing to the average, keeping the data rank full [1]” would be fine. If preprocessing was done in a way that preserved rank, this should be made explicit by stating something along the lines of what I mentioned above. Please refer to the paper and engage with it directly.[1] https://doi.org/10.3389/frsip.2023.1064138
Reply to Comment 4:
We thank the reviewer for pointing out the need for a more detailed description of the ICA implementation and referencing strategy. In response, we have significantly expanded the methodological explanation to clarify the role and implementation of ICA in our framework.
Specifically, we would like to emphasize that ICA was not used as a general preprocessing step, but instead as a modeling tool embedded within the proposed PNDA-MMNet framework. In this context, ICA serves to estimate and subtract the measurement noise term from the observed EEG signals , as part of the state estimation process governed by the linear dynamic system model introduced in Section 2.3. This is crucial for accurately identifying the latent brain state , which is central to our method.
To address the reviewer’s concerns, we have added the following key clarifications in the revised manuscript:
ICA was implemented in MATLAB using an in-house automated algorithm, and was applied in a model-driven rather than preprocessing context, aiming to extract non-neural noise components (e.g., ocular, muscular) from the measurement term EEG signals were re-referenced using the common average reference (CAR). To preserve the full rank of the data matrix and avoid introducing ghost independent components, we retained the original reference electrode during re-referencing, following the recommendation by Kim et al. [26].
These clarifications and methodological details have been added in Section “E. Process Noise Dynamic Adaptation Based Midstream Mesoscopic Network,” where the ICA-based noise removal is first introduced
We believe these revisions address the reviewer’s concerns regarding both the role of ICA in our framework and the data integrity during decomposition.
Comment 5:
The paper lacks a description of the statistical analysis plan, including how the methods were chosen to test the hypotheses. A proper statistical approach should be designed to evaluate whether the observed results support the authors' claims, not just report descriptive differences. The methods section should specify the type of tests used, whether the tests were directional, what assumptions were checked, and how the outcomes would be interpreted in favor or against the hypotheses.
Reply to Comment 5:
We appreciate the reviewer’s insightful comment. In response, we have revised the Methods section (see Section F: Statistical Assessment of Brain State Transition Weights Between NFT and Resting States with FDR Correction) to include a detailed description of the statistical analysis plan. Specifically, we clarified that two-tailed independent-samples t-tests were used to evaluate differences in brain state transition weights and decoding performance metrics between conditions. We have also stated that assumptions of normality and homogeneity of variance were assessed using Shapiro–Wilk and Levene’s tests, respectively. To correct for multiple comparisons, the Benjamini–Hochberg procedure was applied, with statistical significance determined at q < 0.05. The Results section was updated accordingly to reflect these analyses and interpretations in support of our hypotheses. These revisions address the reviewer’s concern regarding the statistical rigor and hypothesis testing approach.
Comment 6:
The authors compare several models, but make no mention of multiple comparison correction. This is a basic statistical requirement when evaluating several classifiers, especially when claiming performance differences.
Reply to Comment 6:
Thank you for your valuable comment. We agree that multiple comparison correction is important when comparing several models. In response, we have clarified our statistical procedures in the revised manuscript. Specifically, we added a new subsection in the Methods (Section F) to describe the use of FDR correction, and we updated the Results section (Section 3.2) to explicitly state that pairwise comparisons of model performance were conducted with FDR correction applied (q < 0.05). These revisions ensure that our performance comparisons are statistically valid.
Comment 7:
The discussion feels overly shallow. It mostly restates the results without critically interpreting them or linking them back to the broader literature. There is little reflection on how well the findings support the original aims, and no serious consideration of limitations or alternative explanations. Main claims are left ungrounded, and the discussion does not add much beyond what is already apparent in the results. This section needs to be expanded to offer real insight and connect the findings to existing knowledge.
Reply to Comment 7:
Thank you very much for your insightful and constructive comments regarding the Discussion section. In response, we have substantially revised and expanded the Discussion to provide deeper analysis and stronger connections to existing literature. Specifically, we have added and enhanced the following parts:
Section A: We discuss the robustness of our model to EEG heterogeneity and spatial resolution limitations, supported by relevant literature to demonstrate its applicability.
Section B: We improve the interpretability of the model by linking the findings to neurophysiological mechanisms, thereby strengthening the biological plausibility of our results.
Section C: We elaborate on the dynamic brain state changes observed in PTSD during neurofeedback training, highlighting the clinical implications revealed by our approach.
Section D: We include a comprehensive limitations section addressing sample size, methodological constraints, and potential future directions, reflecting a critical and balanced perspective.
We believe these revisions have significantly deepened the discussion and enhanced its rigor and relevance. We appreciate your valuable feedback, which has helped improve the manuscript substantially.

Reviewer 2 Report
Comments and Suggestions for Authors
The methods compared in the study (e.g., CNN, LSTM, ENS, and KNN) are traditional models and do not include the latest deep learning architectures. It is suggested to supplement the comparative experiments with more advanced models to fully validate the superiority of PNDA-MMNet.
The paper mentions that Ak is calculated based on the linear dynamic system model, but it does not provide the specific optimization algorithm. Other solution procedures are suggested.
Specific parameter settings are suggested for the supplementary models.
Author Response
Response to Reviewer 2 Comments
Summary:We sincerely appreciate your valuable comments and the time you devoted to reviewing our manuscript. We have carefully addressed each of your suggestions and revised the manuscript accordingly. All changes are highlighted in red in the revised version. We hope that our responses and modifications will meet your approval.
Comment 1:
The methods compared in the study (e.g., CNN, LSTM, ENS, and KNN) are traditional models and do not include the latest deep learning architectures. It is suggested to supplement the comparative experiments with more advanced models to fully validate the superiority of PNDA-MMNet.
Reply to Comment 1:
We sincerely thank the reviewer for raising this important point. We agree that comparing the proposed PNDA-MMNet with more recent deep learning models such as Transformers and Graph Neural Networks (GNNs) would, in principle, further validate its superiority.
However, the choice of baseline methods in our study was guided by both methodological constraints and research objectives.
From a methodological standpoint, advanced deep learning models—such as Transformer-based architectures and GNNs—often require large-scale, stable datasets with minimal variability to perform optimally. Our dataset, by contrast, is characterized by limited training samples, short EEG time sequences, and high inter-subject/session variability, which are inherent to neurofeedback training (NFT) paradigms involving clinical populations. In preliminary experiments, these advanced models underperformed or failed to converge reliably, likely due to overfitting and their sensitivity to unstable signal distributions. Therefore, we selected representative and widely benchmarked models (CNN, LSTM, ENS, and KNN) that are known to be more robust under such challenging data conditions.
More importantly, our study is not solely focused on maximizing classification performance. As explained in our response to another reviewer’s comment, we emphasize process evaluation over outcome validation. That is, our aim is not to demonstrate that improved classification accuracy directly benefits PTSD patients, but rather to decode and interpret brain state transitions throughout the NFT process—a component often overlooked in existing research. In this context, we intentionally selected interpretable and well-established baseline models to serve as meaningful comparisons. Using highly complex and less interpretable models might improve accuracy marginally, but would hinder our ability to link decoding results to neurophysiological mechanisms, thereby weakening the translational relevance of the model.
We have now clarified this rationale more explicitly in the revised manuscript (see Section 3.2). We also acknowledge the value of advanced architectures, and plan to include them in future work where larger datasets and better-controlled conditions may allow for a more informative comparison.
Comment 2:
The paper mentions that Ak is calculated based on the linear dynamic system model, but it does not provide the specific optimization algorithm. Other solution procedures are suggested.
Reply to Comment 2:
Thank you very much for your insightful comment. We appreciate the opportunity to clarify the optimization procedure used to compute the state transition matrix .
In our study, is estimated through a multi-stage optimization pipeline, as detailed in Section 2.B,Section 2.E and illustrated in Figure 3(d). Specifically, the estimation follows a state-space modeling strategy involving:
(1) Solving the discrete-time linear dynamic system equation to obtain a run-specific for each NFT segment;
(2) Temporal alignment across sessions using a Markov Chain-based matching algorithm to identify consistent patterns across the mesoscopic time scale;
(3) Class-level alignment by minimizing a classification loss function in a deep learning framework to ensure cross-subject comparability of matrices.This design enables us to simultaneously capture intra-session dynamics and inter-session commonalities in a physiologically meaningful way.
The complete implementation details are reproducible and supported by the mathematical process and model diagram in Figure 3.
As for the suggestion to consider alternative solution procedures (e.g., nonlinear optimization), we fully acknowledge their potential value—especially for capturing subject-specific variations in brain state transitions. However, the focus of the present study is to explore the shared mesoscopic dynamics underlying the neurofeedback training process across PTSD patients, rather than individual-specific modeling. This is also explicitly stated in Assumption 1, which assumes a consistent dynamic structure (i.e., constant ) within each session to model the common impact of NFT on brain state transitions.
We agree that non-linear or individualized modeling approaches could provide deeper insights into personalized NFT effects, and we see this as an exciting direction for future work. However, such methods would address a different research question from that of the current study, which prioritizes decoding shared neural mechanisms of NFT - induced change.
Comment 3:
Specific parameter settings are suggested for the supplementary models.
Reply to Comment 3:
Thank you for your valuable suggestion regarding the specific parameter settings for the supplementary models. In response to your comment, we have added a new Appendix A1 that outlines the detailed parameter settings for these models. Additionally, we have referred to Appendix A1 in Section 3.2 of the Results section to ensure that the parameters are clearly specified and accessible for readers.
We hope this addition addresses your concerns. Please let us know if any further clarification is required.

Reviewer 3 Report
Comments and Suggestions for Authors
The manuscript introduces PNDA-MMNet, a discrete‐time linear dynamic model with process-noise adaptive Kalman updates that tracks within-session EEG connectivity changes during neurofeedback training for PTSD. On 46 NFT–rest runs from 26 patients, the model attains 0.74 ± 0.12 accuracy, AUC 0.84, outperforming CNN/LSTM/ENS/KNN baselines. The manuscript is well written and the results convincingly support the claims. Here are some minor suggestions before publication.
- Model hyperparameters (learning rate, regularisation, early-stopping criterion) are missing.
- The p-values reported for hundreds of edge weights risk Type-I error. For multiple comparisons, an appropriate correction (e.g., FDR at q < 0.05) is needed.
- The model's interpretability could be improved. It is suggested to clarify how specific PNDA-MMNet parameters map onto neurophysiology.
Author Response
Response to Reviewer 3 Comments
Summary:We sincerely appreciate your valuable comments and the time you devoted to reviewing our manuscript. We have carefully addressed each of your suggestions and revised the manuscript accordingly. All changes are highlighted in red in the revised version. We hope that our responses and modifications will meet your approval.
Comment 1:
Model hyperparameters (learning rate, regularisation, early-stopping criterion) are missing.
Reply to Comment 1:
Thank you for your valuable feedback regarding the missing model hyperparameters. In response to your comment, we have added a table (Appendix A1) that provides the specific settings for the model's hyperparameters, including the learning rate, regularization, and early-stopping criterion. We have also referenced this table in the manuscript to ensure that the hyperparameter details are clearly accessible to the readers.
We hope this revision addresses your concern. Please let us know if further clarification is needed.
Comment 2:
The p-values reported for hundreds of edge weights risk Type-I error. For multiple comparisons, an appropriate correction (e.g., FDR at q < 0.05) is needed.
Reply to Comment 2 :
Thank you for pointing out the risk of inflated Type-I error due to multiple comparisons. In response, we have applied false discovery rate (FDR) correction using the Benjamini-Hochberg procedure across all connectivity comparisons, with a threshold of q < 0.05. We have updated and highlighted the Methods section (Section 2.E),Results section in red.
Comment 3:
The model's interpretability could be improved. It is suggested to clarify how specific PNDA-MMNet parameters map onto neurophysiology.
Reply to Comment 3:
We thank the reviewer for highlighting the importance of model interpretability. To address this, we have revised the Discussion section (now including a new paragraph under the subsection “Model Interpretability and Mapping to Neurophysiology”) to clarify how specific PNDA-MMNet parameters relate to neurophysiological processes.
In particular, we describe how the observation matrix() integrates region-specific functional connectivity derived from EEG, how the state transition matrix () captures NFT-induced modulation in brain dynamics, and how the dynamically adapted process noise () reflects endogenous fluctuations across trials. We further interpret these features in the context of known PTSD neurobiology, including motor and occipital cortical suppression associated with hyperarousal and visual processing abnormalities.
We believe these clarifications strengthen the physiological grounding of our model and enhance its interpretability for both clinical and computational audiences.

Reviewer 4 Report
Comments and Suggestions for Authors
This is a technical paper, with insufficient explanation of terms for the non-specialist reader. Some references (e.g., [29]) do not seem particularly relevant, and do not help the reader to learn about the methods used.
The changes in connectivity described in the Abstract and the features extracted are presented with insufficient commentary, so it is hard for the reader to judge how important they are in the context of the study.
Some methods are presented in insufficient detail. For example, what software was used for ICA (line 196) and what questionnaires were used (line 112). And which statistical tests were conducted, using what software (lines 255-8).
This is a small study (N=26), yet includes quite a number of variables (such as the use of different EEG headsets). Should additional statistical methods be used to strengthen the findings?
What is the meaning of ‘row’ (line 102)? Is this a ‘row’ of electrodes/channels, or a row in a matrix?
Lines 131-166 are difficult to follow. Please provide additional explanation in less technical language.
References require reformatting for consistency, and to follow Journal guidelines. Spaces are missing, brackets need to be removed in some cases, accents should be corrected, volume and issue numbers should be included. Author names are incorrectly presented (e.g., in Ref 36).
Fig 1. Duration of T3 could be included.
Line 141. Justify ‘assumed to be identical.’ In this study it seems that a lot of assumptions have been made about non-heterogeneity.
Line 145. ‘invariant’ – but 2 different devices were used; were data from the same channels recorded with each?
Line 346. ‘they struggle to eliminate the heterogeneity introduced by the experiment’. In comparison, the methods used in this paper do not? But mainly because some assumptions were made about non-heterogeneity? And because ‘The model assumes a discrete linear dynamic system, which may not capture all nonlinearities in brain activity’ (lines 377-8).
Line 225. Why were these ratios (7:1:2) selected? Justify.
Line 228-9. Sentence is missing some characters.
Line 359. ‘radiantly’ – should this be ‘radially’?
Line 381. ‘complexity of the model’. It might interest the reader to know how much time it took to obtain the results for this relatively small dataset, and how the time taken would change with larger datasets.
Were datasets with different numbers of channels also explored?
Author Response
Response to Reviewer 4 Comments
Summary:We sincerely appreciate your valuable comments and the time you devoted to reviewing our manuscript. We have carefully addressed each of your suggestions and revised the manuscript accordingly. All changes are highlighted in red in the revised version. We hope that our responses and modifications will meet your approval.
Comment 1:
This is a technical paper, with insufficient explanation of terms for the non-specialist reader.
Reply to Comment 1:
Thank you for your valuable suggestion. In the revised manuscript, we have thoroughly reviewed the entire text to identify and clarify technical terms. Specifically, we have:
(1) Provided full names alongside abbreviations (e.g., EEG, NFT, ICA, PNDA-MMNet) upon their first appearance in the main text to improve accessibility for non-specialist readers;
(2) Added a glossary (Table 2) summarizing all key custom-defined brain region and network abbreviations used throughout the manuscript;
(3) Supplemented explanations in key sections (e.g., Abstract, Methods 2.E) to clarify specialized terms such as “mesoscale network,” “process noise,” and “midstream network.”
We hope these revisions have enhanced the readability and accessibility of the paper. Please let us know if further clarifications are needed.
Comment 2:
Some references (e.g., [29]) do not seem particularly relevant, and do not help the reader to learn about the methods used.
Reply to Comment 2:
Thank you for your insightful comment. We would like to clarify that the original Reference [29]—Nakano et al. (2016)—was initially intended to support the use of t-distributed stochastic neighbor embedding (t-SNE) in our dimensionality reduction analysis (Figure 4). However, due to a citation placement error, it was mistakenly included in the section describing comparative machine learning models (e.g., CNN, LSTM, KNN), which led to confusion regarding its relevance.
In the revised manuscript, we have corrected this citation mistake. Specifically, we removed the misplaced citation of Nakano et al. and replaced it with the authoritative and widely cited reference for t-SNE:
Laurens V D M , Hinton G .Visualizing Data using t-SNE[J].Journal of Machine Learning Research, 2008, 9(2605):2579-2605.
This ensures that the reference accurately supports the use of t-SNE and is no longer misattributed to unrelated methods. We appreciate the reviewer’s careful attention, which helped us improve the accuracy and clarity of our manuscript.
Comment 3:
The changes in connectivity described in the Abstract and the features extracted are presented with insufficient commentary, so it is hard for the reader to judge how important they are in the context of the study.
Reply to Comment 3:
Thank you for this important comment. We agree that clear interpretation of the observed connectivity changes and extracted features is essential for understanding their significance.
In the revised manuscript, we have already provided a dedicated section in the Discussion (Section 4.C: Dynamic Brain State Changes in PTSD During NFT Revealed by PNDA-MMNet) where we analyze these features in depth. Specifically, we describe:
The directional suppression of connectivity within the motor cortex (especially in the right hemisphere);
A radial pattern of inhibition in the left occipital cortex;
The preservation of connectivity between motor and occipital regions;
The correspondence of these findings with known neurobiological models of PTSD.
These observations offer insight into how NFT may modulate hyperactivity in key cortical areas and reflect underlying therapeutic mechanisms.
To make this more accessible to readers at a glance, we have also added a brief interpretive statement in the Abstract highlighting the relevance of the connectivity changes to PTSD-related dysfunction and NFT mechanisms.
We appreciate the reviewer’s suggestion, which helped improve the clarity and contextual significance of our findings.
Comment 4:
Some methods are presented in insufficient detail. For example, what software was used for ICA (line 196) and what questionnaires were used (line 112). And which statistical tests were conducted, using what software (lines 255-8).
Reply to Comment 4:
Thank you very much for your careful review and valuable comments. In response to your concern regarding insufficient detail in the Methods section, we have made the following revisions:
Details regarding the software used for Independent Component Analysis (ICA) have been added in Section E of the Methods.
Our study includes two types of data: public data and self-collected data. The public dataset has been fully described in Reference [5]. Since our study does not involve pre-post comparison assessments, we did not reiterate these details in the main text. For the self-collected data, demographic information and inclusion/exclusion criteria have been supplemented in Section A of the Methods and Appendix A1.
We have added a new Section F in the Methods to comprehensively describe the statistical tests.
We hope these revisions adequately address your concerns. Thank you again for your constructive feedback, which has helped improve the quality of our manuscript.
Comment 5:
This is a small study (N=26), yet includes quite a number of variables (such as the use of different EEG headsets). Should additional statistical methods be used to strengthen the findings?
Reply to Comment 5:
Thank you very much for your careful review and valuable comments. Regarding your concern about the small sample size and multiple variables, and whether additional statistical methods should be applied to strengthen the findings, we would like to clarify the following:
First, although the sample size in this study is relatively small (N=26), such a sample size is already considered quite challenging and representative in the field of post-traumatic stress disorder (PTSD) research, given the inherent difficulties in patient recruitment and data collection.
Second, to address the issues of limited sample size, multiple variables, and data heterogeneity, we did not rely on traditional statistical methods. Instead, we developed a specialized deep learning model, PNDA-MMNet, which models data at the session level, extracts dynamic features from each run, and systematically compares its performance with typical machine learning and deep learning frameworks. This approach effectively mitigates the challenges posed by limited sample size and complex data characteristics.
Furthermore, in response to your concerns, we have added a new section in the revised manuscript titled “Robustness of PNDA-MMNet to EEG Heterogeneity and Limited Spatial Resolution,” which discusses in detail how our model addresses the issues of variable heterogeneity, data variability, and potential overfitting.
We believe these improvements enhance the robustness and credibility of our findings. We sincerely appreciate your insightful suggestions.
Comment 6:
What is the meaning of ‘row’ (line 102)? Is this a ‘row’ of electrodes/channels, or a row in a matrix?
Reply to Comment 6:
Thank you for your insightful comment. In the manuscript, the term "row" refers primarily to an EEG channel or electrode label. Each "row" corresponds to a specific EEG electrode. At the same time, in the data matrix representation, this "row" corresponds to one dimension (i.e., a single row) that contains the time-series data recorded from that particular channel. Thus, the term "row" denotes both the EEG electrode/channel and its associated data vector within the matrix.
Comment 7:
Lines 131-166 are difficult to follow. Please provide additional explanation in less technical language.
Reply to Comment 7:
Thank you for this valuable comment. We fully agree that the use of control theory concepts in modeling neural feedback training (NFT) processes may pose a comprehension challenge for readers unfamiliar with this field. However, the introduction of linear discrete dynamic system modeling—derived from control science—is a key methodological innovation of this study. Rather than reducing the technical depth in the main text, we have chosen to preserve the formalism and provide an accessible explanation in a supplementary paragraph. This approach ensures both academic rigor and reader accessibility.
Accordingly, we have added a concise explanatory passage directly below Section 2.E to help readers understand the logic of the model from a non-technical perspective.
Comment 8:
References require reformatting for consistency, and to follow Journal guidelines. Spaces are missing, brackets need to be removed in some cases, accents should be corrected, volume and issue numbers should be included. Author names are incorrectly presented (e.g., in Ref 36).
Reply to Comment 8:
Thank you for pointing this out. We have carefully revised the entire reference list to comply with the Sensors reference formatting guidelines, including proper citation order, author formatting, journal abbreviations, and DOI inclusion where applicable.
Comment 9:
Fig 1. Duration of T3 could be included.
Reply to Comment 9:
Thank you for your suggestion. We have added the duration of the T3 phase in Figure 1 to improve clarity. The figure and its legend have been updated accordingly.
Comment 10:
Line 141. Justify ‘assumed to be identical.’ In this study it seems that a lot of assumptions have been made about non-heterogeneity.
Reply to Comment 10:
Thank you very much for the reviewer’s valuable comment. The assumption of identical state transition matrices is grounded in the primary objective of this study, which is to extract the common neural mechanisms underlying NFT processes across different PTSD patients. Therefore, we assume that the state transition matrices across different runs are consistent, aiming to identify these shared features. It is important to emphasize that this assumption does not imply neglecting the heterogeneity caused by inter-subject and inter-session variability. On the contrary, our model is specifically designed to overcome such heterogeneity in order to reliably extract the consistent neural patterns. We have added a dedicated section in the Discussion (Section A) to elaborate further on this point.
Moreover, the formulation of Assumption 1 does not preclude extending the model to nonlinear cases. In other words, it is possible that the brain changes induced by NFT vary among individuals if nonlinear effects are considered(if ,the model can still be solved by other algorithms). This represents a distinct research direction focusing on personalized NFT modeling. However, such an approach emphasizes individual-specific characterizations rather than the common underlying mechanisms of the NFT process, which is the main focus of the present study for process evaluation.
Comment 11:
Line 145. ‘invariant’ – but 2 different devices were used; were data from the same channels recorded with each?
Reply to Comment 11:
We sincerely appreciate the reviewer’s insightful comment. Our response is as follows:
(1)The term “invariant” refers to the control input matrix B, which represents external inputs introduced into a stable control system. It does not refer to the EEG signal itself. In our model, the EEG signal is represented by the observation matrix Zk , while the actual brain state is denoted as xk . These are related via the observation matrix H, and the recorded signal also includes measurement noise vk. For instance, in a system continuously acquiring resting-state EEG, zk reflects the recorded signals without external interference. If the researcher introduces a stimulus (e.g., visual or auditory), this can be considered as an external control input B*uk affecting the system. However, in our study, once the NFT procedure was initialized, no further external stimuli were introduced. Therefore, we consider B=0 throughout, meaning the system is “invariant” with respect to external control inputs during the NFT process.
(2) Regarding the use of two EEG devices, it is true that the public dataset was acquired with one device (with a specific number of channels), while our own recordings used a 32-channel system. However, during preprocessing, we standardized the data across devices by normalizing signal magnitudes and mapping the channels to a common set of scalp regions. This ensured consistency in both dimensionality and spatial coverage. We did not directly input raw EEG data from different devices into the model. The detailed data alignment and normalization procedures are described in Section D of the Methods.
Comment 12:
Line 346. ‘they struggle to eliminate the heterogeneity introduced by the experiment’. In comparison, the methods used in this paper do not? But mainly because some assumptions were made about non-heterogeneity? And because ‘The model assumes a discrete linear dynamic system, which may not capture all nonlinearities in brain activity’ (lines 377-8).
Reply to Comment 12:
We appreciate the reviewer’s thoughtful comment, which is closely related to Comment 10. In this study, our primary objective was to investigate the generalizable effects of neurofeedback training (NFT) on brain activity in PTSD patients across subjects and sessions. To achieve this, we deliberately designed the model to extract features that reflect common patterns rather than subject-specific variability. Accordingly, we assumed consistency and linearity in this part of the system. However, this does not imply an assumption that there is no heterogeneity across subjects or sessions; rather, the model is built to extract stable features under inherently heterogeneous conditions.
We fully agree that nonlinear dynamics play a significant role in brain function and are worthy of further investigation. However, the core aim of this study is to identify stable and reproducible neural effects of NFT, which requires a modeling framework that prioritizes interpretability and generalizability. While our framework can be extended to incorporate nonlinear components in future work, such modifications would be more suitable for studying individual differences or subject-specific responses, rather than for capturing the shared neural mechanisms underlying NFT intervention.
Comment 13:
Line 225. Why were these ratios (7:1:2) selected? Justify.
Reply to Comment 13:
We thank the reviewer for pointing this out. The dataset was split into training, validation, and test sets using a ratio of 7:1:2, which is a commonly adopted convention in machine learning and deep learning research. This split ensures that a sufficiently large portion of the data is allocated for model training (70%), while still retaining separate sets for validation (10%)—to fine-tune hyperparameters and avoid overfitting—and testing (20%)—to evaluate the model's generalization performance. Such ratios have been widely used in the literature as a balance between training sufficiency and robust performance evaluation.
Comment 14:
Line 228-9. Sentence is missing some characters.
Reply to Comment 14:
We appreciate the reviewer’s careful reading. The incomplete sentence has been revised.
Comment 15:
Line 359. ‘radiantly’ – should this be ‘radially’?
Reply to Comment 15:
Thank you for pointing this out. We agree with the reviewer’s suggestion. The word “radiantly” was indeed a typographical error and has been corrected to “radially” in the revised manuscript. The corrected sentence now reads:
“This suggests that the degree of inhibition in the LOC area during NFT increases radially from the midline toward the periphery of the occipital region.”
This revision improves both the accuracy and clarity of the description.
Comment 16:
Line 381. ‘complexity of the model’. It might interest the reader to know how much time it took to obtain the results for this relatively small dataset, and how the time taken would change with larger datasets.
Reply to Comment 16:
We thank the reviewer for the valuable suggestion regarding the computational time of our model. Our work focuses on neurofeedback training process evaluation, which essentially operates as an online brain-computer interface system. Therefore, real-time performance is critical for clinical usability.
Currently, after training, the model’s inference time on our relatively small dataset is on the order of 10 seconds per evaluation. While this meets preliminary experimental needs, it is recognized that practical NFT systems require faster, ideally second-level or near-instantaneous responses to ensure seamless feedback and effective clinical intervention.
Hence, we acknowledge that the model’s current complexity could limit its real-time applicability in clinical settings without further optimization. Future work will focus on optimizing the model’s computational efficiency and exploring potential simplifications to better accommodate larger and more diverse patient populations while maintaining real-time performance.
We have revised the manuscript accordingly to reflect these points (see Section 4.D Limitation).
Comment 17:
Were datasets with different numbers of channels also explored?
Reply to Comment 17:
We appreciate the reviewer’s insightful question regarding datasets with different numbers of channels. In our study, we utilized EEG data from publicly available datasets, which include two types of channel configurations, as well as a self-collected dataset from our hospital with a single channel configuration. To ensure consistency in model training, all data were normalized to a common channel framework.
If the reviewer is inquiring about testing different channel layouts, we regret to note that the field of PTSD neurofeedback process evaluation remains relatively underexplored, and our ongoing hospital-based experiments have not yet accumulated sufficient data to thoroughly investigate the effects of various channel arrangements. We acknowledge the importance of this aspect and plan to extend our work by incorporating datasets with diverse channel numbers and layouts in future studies to comprehensively evaluate the generalizability and robustness of our approach.
We thank the reviewer again for this valuable suggestion.

Round 2
Reviewer 1 Report
Comments and Suggestions for Authors
The authors successfully addressed the issues.
Author Response
We sincerely thank the reviewer for the positive evaluation and kind comments. We are pleased that our revisions have addressed the previous concerns. We greatly appreciate your time and support.
Reviewer 4 Report
Comments and Suggestions for Authors
Thank you for your careful revisions. Your paper is now much more readable and informative.
Table 1 is particularly useful, and table 2 is a useful summary. The explicit description of statistical tests used is good and most helpful.
Line 317: Give some explanation of ’q’ for FDR
Lines 260, 274, 345, 389: Check punctuation
Footnotes to Table A1 – I cannot see the numbers in the Table itself.
Author Response
Response to Reviewer 4 Comments
Summary:We sincerely appreciate your valuable and meticulous comments as well as the time and effort you devoted to reviewing our manuscript for the second time. Your detailed suggestions have greatly helped us improve the clarity and quality of our work. We have carefully addressed each of your concerns point-by-point and revised the manuscript accordingly. In particular, we have corrected all punctuation issues you identified, clarified the explanation of the FDR-related notation (q value), and improved the linkage between the footnotes and the corresponding table entries in Table A1. All changes are clearly highlighted in green in the revised manuscript. We hope that our responses and the revised version meet your expectations and approval.
Comment 1:
Line 317: Give some explanation of ’q’ for FDR
Reply to Comment 1:
Thank you for your comment. We have clarified the meaning of ‘q’ in the revised manuscript (Line 317), now stating that q refers to the FDR-adjusted p-value, which reflects the expected proportion of false positives among the results deemed significant after correction for multiple comparisons.
Comment 2:
Lines 260, 274, 345, 389: Check punctuation
Reply to Comment 2:
Thank you for your careful reading and helpful suggestions regarding punctuation (Lines 260, 274, 345, and 389). We have carefully reviewed these sentences and corrected the punctuation issues, including misplaced commas and spacing. In addition, we have slightly revised the sentence structures to improve clarity and readability. The revised text can be found in the corresponding lines of the manuscript.
Comment 3:
Footnotes to Table A1 – I cannot see the numbers in the Table itself.
Reply to Comment 3:
Thank you for pointing this out. We have revised Table A1 to ensure that all footnotes are clearly linked to the corresponding values in the table using superscript markers. The footnote numbering is now visible and consistent with the references in the table.